# Cyanidin-3-O-Glucoside Induces the Apoptosis of Human Gastric Cancer MKN-45 Cells through ROS-Mediated Signaling Pathways

**DOI:** 10.3390/molecules28020652

**Published:** 2023-01-09

**Authors:** Wei Sun, Nai-Dan Zhang, Tong Zhang, Yan-Nan Li, Hui Xue, Jing-Long Cao, Wen-Shuang Hou, Jian Liu, Ying Wang, Cheng-Hao Jin

**Affiliations:** 1Department of Food Science and Engineering, College of Food Science, Heilongjiang Bayi Agricultural University, Daqing 163319, China; 2Department of Biochemistry and Molecular Biology, College of Life Science & Technology, Heilongjiang Bayi Agricultural University, Daqing 163319, China; 3National Coarse Cereals Engineering Research Center, Daqing 163319, China

**Keywords:** cyanidin-3-O-glucoside, gastric cancer cells, apoptosis, cell cycle, migration

## Abstract

Cyanidin-3-O-glucoside (C3G), an active ingredient in anthocyanins, mainly exists in dark cereals. C3G was investigated for its effect on human gastric cancer (GC) cells, together with its molecular mechanism. The CCK-8 assay results showed that C3G had significant antiproliferative effects on GC cells, but it had little effect on normal cells. Western blot and flow cytometry results showed that C3G regulated the reduction of mitochondrial membrane potential and arrested the cell cycle in the G2/M phase through the AKT signaling pathway, causing the cells to undergo apoptosis. Additionally, in MKN-45 cells, C3G markedly raised intracellular reactive oxygen species (ROS) levels. The wound healing assay and Transwell assay results showed that MKN-45 cell migration was significantly inhibited. Western blot results showed that the expression of E-cadherin protein was upregulated and the expressions of β-catenin, N-cadherin, and Vimentin were downregulated. Additionally, following N-acetylcysteine treatment, the expression levels of these proteins were reduced. In conclusion, C3G caused MKN-45 cells to undergo apoptosis; arrested the cell cycle in the G2/M phase; hindered cell migration; and activated the MAPK, STAT3, and NF-κB signaling pathways, by inducing an increase in ROS levels. Thus, C3G may be a promising new medication for the treatment of GC.

## 1. Introduction

Gastric cancer (GC) is one of the most common cancers worldwide, affecting more than 1.2 million people each year, while its mortality rate ranks third in cancer mortality, causing great harm to human health [1,2]. 5-Fluorouracil and cisplatin are often used for the treatment of GC, but they cause significant adverse reactions and are costly [3]. Therefore, it is urgent to find a low toxicity and effective drug. GC is closely related to diet, and the long-term intake of fruits, vegetables, and soy products can reduce the risk of GC, possibly because they are rich in anthocyanins [4,5]. Anthocyanins have free radical scavenging, antioxidant, and inhibitory effects on cancer cell proliferation.

Apoptosis is a highly efficient way to kill malignant tumor cells, is necessary for homeostasis, and is strictly controlled by multiple genes [6]. The Bcl-2 family of proteins, specifically the anti-apoptotic protein (Bcl-2) and pro-apoptotic protein (Bad), as well as the terminal shearing enzyme caspase-3, are essential parts of the mechanism of cell apoptosis [7]. Reactive oxygen species (ROS) are formed by the reduction of molecular oxygen by a single electron. ROS can promote signal transduction between cells at low levels, but in excessive doses, they can trigger cell apoptosis [8]. ROS are involved in the activation of many signaling pathways, for example, signal transducer and activator of transcription 3 (STAT3), mitogen-activated protein kinase (MAPK), and nuclear factor kappa B (NF-κB) [9,10].

Cyanidin-3-O-glucoside (C3G) is a class of water-soluble natural compounds with antioxidant activity and mainly exist in dark grains, such as black beans, black rice, and purple sweet potato. It is one of the most common and most studied anthocyanins [11]. C3G can induce cell apoptosis in colon cancer and glioblastoma, and it was also found that C3G can reduce ROS levels by regulating the Nrf2 signaling pathway [12,13,14,15,16]. C3G can prevent HeLa cell cycle arrest in G1 phase and inhibit the cell migration of human retinal endothelial cells [17,18]. C3G can inhibit breast cancer angiogenesis through the STAT3 signaling pathway [19]; however, C3G’s inhibition of human GC cells remains elusive in terms of its specific molecular mechanism.

This study investigated the antitumor effects of C3G on the apoptosis, ROS accumulation, cell cycle arrest, and migration inhibition of GC MKN-45 cells. In addition, we elucidated the roles of the MAPK, STAT3, and NF-κB signaling pathways in the C3G-induced apoptosis of MKN-45 cells.

## 2. Results

### 2.1. C3G Inhibits the Growth of GC Cells

Cis-diaminodichloroplatin (DDP) is a widely used drug in the treatment of cancer and can inhibit the DNA replication process of cancer cells and damage the structure of the cell membrane, showing a strong broad-spectrum anticancer effect. It is often used in malignant tumors, such as gastric cancer, ovarian cancer, breast cancer, and brain cancer. In this study, DDP was used as the positive control, to verify the killing effect of C3G and DDP on gastric cancer cells and the toxic and side effects on normal cells. As shown in Figure 1A,B, C3G inhibited GC cell growth in a dose-dependent and time-dependent manner, and the effects of concentration and time were significantly higher than that of DDP. IC_50_ values are shown in Table 1. In addition, the effects of C3G and DDP on the human normal cells THLE-2, 293-T, IMR-90, and GES-1 were detected using the same method. According to Figure 1C,D, the adverse effects of C3G were significantly lower than those of DDP, which had little impact on human health. According to the above results, as compared to the other 11 GC cell lines, MKN-45 cells were more sensitive to C3G and showed the lowest side effects, with IC_50_ values reaching 87 μM at 24 h. These results meant that MKN-45 cells were selected for subsequent verification, and after 36 h of C3G treatment, very few MKN-45 cells survived, so 87 μM C3G was used to terminate the follow-up treatment at 24 h.

### 2.2. C3G Induces the Apoptosis of MKN-45 Cells

As seen in Figure 2, the fluorescence intensity of MKN-45 cells was enhanced after 24 h of C3G treatment compared with the DDP treatment, which showed apoptotic effects. Moreover, cell rounding and cell fragmentation were observed under bright-field microscopy. As shown in Figure 3A, after C3G treatment, the apoptosis rate of MKN-45 cells increased from 0.17% to 56.26% from 0 to 24 h. In addition, after the C3G treatment shifted from the first to fourth quadrant, the red/green fluorescence ratio gradually decreased and the MMP (mitochondrial membrane potential) depolarized, indicating that the MMP of MKN-45 cells decreased over time and that apoptosis occurred (Figure 3B). C3G significantly increased Bad, Cyto C, cle-caspase-3, and cle-PARP, while significantly decreasing the expression of Bcl-2 (Figure 3C). These findings indicate that C3G can induce apoptosis of MKN-45 cells through the mitochondrial apoptosis pathway.

### 2.3. C3G Induces the Apoptosis of MKN-45 Cells through the MAPK/STAT3/NF-κB Signaling Pathway

As seen in Figure 4A, C3G treatment led to p-p38 and p-JNK being increased in MKN-45 cells, while the expression levels of p-ERK, p-STAT3, NF-κB, and I-κB-α were significantly decreased. ERK inhibitor (FR180204), JNK inhibitor (SP600125), and p38 inhibitor (SB203580) were added to the cells for 30 min prior to C3G therapy, to further confirm the connection between MAPK and STAT3. Figure 4B–D shows that JNK and p38 inhibitors reduced the expression levels of p-STAT3 compared to the C3G group. The expression levels of p-STAT3 increased after the ERK inhibitor treatment. These results suggest that after C3G treatment, MAPK participates in regulating the STAT3 signaling pathway and induces the apoptosis of MKN-45 cells.

### 2.4. C3G Induces the Apoptosis of MKN-45 Cells through ROS-Mediated MAPK/STAT3/NF-κB Signaling

With time, the C3G-treated MKN-45 cells’ ROS levels rose (Figure 5A), but decreased with time in GES-1 cells treated with C3G (Figure 5B). To observe the effect of ROS on cell apoptosis, MKN-45 cells were pre-processed with NAC. The apoptosis proportion of cells in the NAC + C3G group was 28.78%, which was much lower than the 53.36% of cells treated with C3G alone (Figure 5C). The histone expression of the NAC + C3G group was significantly inhibited compared with the C3G group (Figure 5D). These results suggest that C3G increases ROS levels and participates in the induction of apoptosis in MKN-45 cells.

### 2.5. C3G Induces MKN-45 Cell Cycle Arrest through AKT Signaling

After C3G treatment for 0–24 h, the number of cells in the G0/G1 phase decreased by 19.64%, and that of cells in the G2/M phase increased by 6.64% (Figure 6A). Western blot results showed the expression levels of AKT, CDK1/2, and cyclin B1 were decreased, while the expression level of p27 was increased (Figure 6B). The G2/M phase of the C3G + NAC group was 3.6% lower than that of the C3G group, indicating that NAC could inhibit cell cycle arrest (Figure 6C). Protein expression was reversed in the C3G + NAC group compared with the C3G group (Figure 6D). These findings indicate that C3G can induce the G2/M phase cycle arrest of MKN-45 cells through the ROS-mediated AKT signaling pathway.

### 2.6. C3G Inhibits MKN-45 Cell Migration

As shown in Figure 7A, MKN-45 cells treated with C3G migrated significantly slower than untreated MKN-45 cells after 3 h, and this difference was highly significant after 12 h. The migration capacity of MKN-45 cells treated with C3G was reduced compared with the control group (Figure 7B). The protein expression levels were analyzed using Western blot. The expression of E-cadherin was increased, while that of β-catenin, N-cadherin, and Vimentin was significantly decreased (Figure 7C). Similarly, pretreatment with NAC significantly inhibited cell migration compared with treatments with C3G alone (Figure 7D). Additionally, compared to the C3G group, the NAC + C3G group’s expression of E-cadherin was much higher, whereas that of β-catenin, N-cadherin, and Vimentin was lower (Figure 7E). These results suggest that CSG can effectively inhibit cell migration and proliferation.

## 3. Discussion

DDP is a highly effective drug for inducing apoptosis of cancer cells and can induce apoptosis of ovarian cancer cells [20]. This study confirmed that DDP can also inhibit the growth of GC cells, but the effect is not as effective as C3G. The reason for this may be that DDP can inhibit the growth of cancer cells by destroying the cell membrane structure, while C3G can directly penetrate cancer cells. Anthocyanins, water-soluble pigments, are widely found in dark plants, fruits, and vegetables. They have anti-inflammatory effects in diabetes and cardiovascular diseases [21,22,23,24,25,26,27,28,29,30], and also inhibit the proliferation of AGS cells in GC [31]. C3G is the main active form of anthocyanin [32], and we explored whether C3G also has the effect of inhibiting GC cells. Our results showed that C3G inhibited growth of GC cells, but had little killing effect on normal human cells. However, unlike a previous study [31], we found that C3G was more sensitive to human GC MKN-45 cells. We speculated that the mechanism of action of C3G was different from that of anthocyanin on GC cells. Subsequently, we continued to study the apoptosis-inducing influence of C3G on MKN-45 cells and determined its detailed molecular mechanism.

Programmed cell death, namely, apoptosis, is an important regulatory mode to maintain body homeostasis [33]. Caspase-3 can cause mitochondria-mediated apoptosis or destroy the MMP [34,35]. C3G had inhibitory effects on cervical cancer, and the level of Bcl-2 protein decreased, which activated cle-caspase-3 [17]. Our study also showed that C3G induced the apoptosis of MKN-45 cells, inhibited the apoptosis-related protein expression level, and decreased the MMP. Consequently, we postulated that the C3G-mediated induction of MKN-45 cell apoptosis may have been due to its regulation of mitochondria-related pathways.

ROS are necessary for many signal transduction pathways and can trigger the physiological pathways of cell apoptosis [36]. *Helicobacter pylori* induces increased activation of ROS, MAPK, and NF-κB in AGS cells, but the ROS levels in AGS cells decrease after anthocyanin pretreatment [30]. Our results showed that the ROS levels of MKN-45 cells increased after C3G treatment, while the ROS levels of human gastric epithelial GES-1 cells decreased after C3G treatment. We believe that C3G has a strong antioxidant activity, but the molecular pathways in normal cells and cancer cells differ from one another, involving different signaling pathways, which needs to be investigated further. C3G can provide effective protection against the adverse effects of ultraviolet B radiation, by regulating the MAPK and NF-κB signaling pathways [14], and can also inhibit breast cancer angiogenesis, by inhibiting the STAT3 pathway [15]. As a natural active ingredient, anthocyanin can block the proliferation of various tumor cells through the terminal differentiation of tumor cells, and further induce the apoptosis of tumor cells through the STAT3 and NF-κB signaling pathways regulated by ROS. At the same time, anthocyanin can eliminate free radicals in the body and generate antioxidant activity, thus achieving the effect of preventing the generation of tumor cells [37,38,39]. In this study, C3G induced the apoptosis of MKN-45 cells by regulating the MAPK, STAT3, and NF-κB signaling pathways, consistently with previous studies.

The cell cycle is tightly regulated and ordered, and small molecule drugs have been widely used to treat cancer by interfering with the cell cycle [40]. Anthocyanins can induce the cycle arrest and apoptosis of oral cancer cells and leukemia cells in the G2/M phase [41,42]. This study further demonstrated that C3G can cause the G2/M phase cell cycle arrest of GC MKN-45 cells, while downregulating the levels of p-AKT, cyclin B1, and CDK 1/2 protein expression and upregulating the levels of p27 protein expression. These data indicate that C3G induces MKN-45 cells to undergo cycle arrest in the G2/M phase, through the induction of AKT signaling, and then induces apoptosis.

Cell migration is one of the fundamental functions of cells and is crucial in tumor metastasis. The epithelial–mesenchymal transition (EMT) plays a significant role in tumor invasion and metastasis, by altering the adhesion molecules produced by cells, while E-cadherin and Vimentin participate in the EMT process [43,44]. C3G can significantly inhibit melanoma cell migration through the AKT signaling pathway [45]. In our study, we also found that C3G inhibits the migration of MKN-45 cells through β-catenin and that regulates the protein expression levels of N-cadherin, E-cadherin, and Vimentin at the molecular level. We postulated that C3G can inhibit the migration of tumor cells through different signaling pathways in different cells.

In this study, C3G induced apoptosis in GC MKN-45 cells via ROS-mediated MAPK, STAT3, and NF-κB signaling pathways (Figure 8). The findings of this investigation can contribute to a better understanding of the molecular mechanism by which C3G induces apoptosis in GC cells, laying the groundwork for future anticancer drug research and development.

## 4. Materials and Methods

### 4.1. Materials

C3G (purity ≥98.0%) was purchased from Herbpurify Co., Ltd. (Chengdu, China). DDP (purity ≥99.0%) was from Solarbio (Beijing, China). AGS, MKN-45, HGC-27, SGC-7901, MKN-74, BGC-823, MKN-28, KATO-3, YCC-1, NCI-N87, MGC-803, and YCC-6 cells were from the American Type Culture Collection (Manassas, VA, USA). THLE-2 immortalized human liver cells were from Shenzhen Haodi Huatuo Biological Technology Co., Ltd. (Shenzhen, China). 293T human embryonic kidney cells, IMR-90 human embryonic lung fibroblasts, and GES-1 human gastric epithelial mucosa cells were from Shanghai Qingqi Biotechnology Development Co., Ltd. (Shanghai, China).

### 4.2. Cell Culture

MKN-45, MKN-28, MKN-74, KATO-3, AGS, HGC-27, NCI-N87, YCC-1, BGC-823, MGC-803, SGC-7901, and YCC-6 human GC cells; and 4 kinds of normal cells THLE-2, 293T, IMR-90, and GES-1 were grown in Dulbecco’s Modified Eagle Medium supplemented with 10% fetal bovine serum (FBS; Gibco, Waltham, MA, USA) and 1% penicillin/streptomycin (Gibco). The ambient temperature was 37 °C and 5% CO_2_.

### 4.3. Cell Viability Assay

A Cell Counting Kit-8 (CCK-8; Solarbio) was used to measure cell viability [46]. DDP control was chosen because of its significant inhibitory effects on cancer cells and widespread clinical usage. GC and normal cells were seeded in 96 well plates (1 × 10^4^ cells/well). C3G and DDP were incubated at various concentrations (30, 60, 90, 120, and 150 µM) and various time points (3, 6, 12, 24, and 36 h). The absorbance (optical density (OD)) was then measured at 450 nm using a Microplate Luminometer (BioTek Instruments, Inc., Winooski, VT, USA), after CCK-8 was introduced to determine cell viability. Calculation of the half maximal inhibitory concentration (IC_50_) was carried out using GraphPad Prism 5.0.

### 4.4. Apoptosis Analyses

The Annexin V-FITC/PI Apoptosis Detection Kit (Solarbio) was used to examine cell apoptosis [47]. In 6 well plates, human GC MKN-45 cells (1 × 10^5^ cells/well) were grown and exposed to 87 μM C3G for 3, 6, 12, and 24 h. Then, 1 mL phosphate-buffered saline (PBS; Hyclone, South Logan, UT, USA), 195 µL binding buffer, 2 µL Annexin V-FITC, and PI were added. Fluorescence intensity was observed with an inverted microscope (Mingmei Optoelectronics Co., Shanghai, China). Additionally, MKN-45 cells were incubated with 95 µL Annexin V binding buffer, 3 µL Annexin V-FITC and 2 µL PI; and a flow cytometer (Schisen Micron Medical Electronics Co., Shanghai, China) was used to determine the rate of apoptotic cells. MKN-45 cells pretreated with N-acetylcysteine (NAC) were treated with C3G and then tested.

### 4.5. MMP Analysis

An MMP Kit (Solarbio) was used to test the MMP [48]. In 6 well plates, MKN-45 cells (1 × 10^5^ cells/well) were grown and exposed to 87 μM C3G for 3, 6, 12, and 24 h and incubated for 30 min in 500 µL JC-1 staining solution. Then, cells were centrifuged at 1450 rpm for 4 min, with the addition of 500 µL JC-1 binding buffer to suspend the cells. The MMP was measured using flow cytometry. Cell apoptosis leads to depolarization of mitochondrial transmembrane potential, and JC-1 is released from mitochondria, its concentration is reduced, and it reverses into a monomer form emitting a green fluorescence. Changes in MMP can be quantitatively detected by detecting the migration of green and red fluorescent cell populations.

### 4.6. Analysis of Cell Cycle Arrest

A Cell Cycle Kit (Solarbio) was used assess to cell cycle arrest [7]. MKN-45 cells were treated with C3G for 3, 6, 12, and 24 h, washed once with cold PBS, and centrifuged for five minutes, with overnight fixation in 70% ethanol at 4 °C. Then they were incubated with 100 µL RNaseA at 37 °C for 30 min, followed by incubation with 400 µL PI at 4 °C for 30 min to avoid light. Cells cycle were detected using flow cytometry. MKN-45 cells pretreated with NAC were treated with C3G and then tested.

### 4.7. Western Blot Analysis

Western blot analysis was used to detect the specific expression levels of protein [49]. MKN-45 cells were treated with 87 μM C3G for 3, 6, 12, and 24 h, to extract protein. Then, 8–15% SDS-PAGE electrophoresis gels with various concentrations were used to separate the proteins and electrotransfer them to nitrocellulose membranes. Membranes were incubated with the following primary antibodies (Santa Cruz Biotechnology, Inc., Dallas, TX, USA) overnight at 4 °C after being blocked in 5% skimmed milk for 2 h; to investigate whether C3G induced GC cell apoptosis through the MAPK/STAT3/NF-κB signaling pathway, arrested cell cycle through AKT signaling pathway, and inhibited cell migration through the β-catenin signaling pathway: α-tubulin, AKT, p-AKT, Bcl-2, Bad, cle-caspase-3, cyclin B1, cle-PARP, Cytochrome C (Cyto C), cyclin-dependent kinase (CDK1/2), ERK, p-ERK, STAT3, p-STAT3, JNK, p-JNK, NF-κB, I-κB-α, p27, p38, p-p38, β-catenin, N-cadherin, E-cadherin, and Vimentin. The membrane was then incubated with secondary antibody (ZSGB-BIO, Inc., Beijing, China) for 2 h. Enhanced chemiluminescence and UVP ChemStudio 515 (Schisen Micron Medical Electronics Co., Ltd.) were used for protein visualization, the content was normalized using ImageJ 1.46R software, and α-tubulin was used as an internal control.

### 4.8. Analyses of ROS Levels

A ROS detection kit (Beyotime Institute Biotechnology, Shanghai, China) was used to analyze the levels of ROS accumulation [50]. The purpose of this study was to investigate the effect of C3G on the ROS levels in GC cells, but C3G has antioxidant properties, so it is necessary to select normal stomach cells for ROS level detection. GES-1 cells are normal gastric mucosal epithelial cells that also grow in the stomach, so we chose GES-1 as the control. In 6 well plates, MKN-45 cells and GES-1 cells (1 × 10^5^ cells/well) were grown and exposed to 87 μM C3G for 3, 6, 12, and 24 h. Cells were incubated with DCFH-DA for 30 min without light, then washed with PBS. Flow cytometry was used to detect the accumulation of ROS.

### 4.9. Analysis of Cell Migration

The migration of C3G to MKN-45 cells was investigated in a cell scratch experiment [51]. In 6 well plates, MKN-45 cells (1 × 10^5^ cells/well) were grown and cultured, to observe the cell morphology. Then, cells were scratched in the petri dish and treated with 87 μM C3G. Under an inverted microscope, cell migration was observed for 3, 6, 12, and 24 h. MKN-45 cells were pretreated with NAC, before treating them with C3G, and then tested. Cell migration was detected using the Transwell method [52]. An MKN-45 cell suspension at a concentration of 1 × 10^5^/mL was prepared in the upper Transwell chamber. When the cell density was 90–95%, and after serum starvation in medium containing 1% FBS, the cells were treated with 87 μM C3G for 3, 6, 12, and 24 h. Transwell chambers were cultured for 24 h with culture medium containing 20% FBS and stained with 0.1% crystal violet. Bright fields were observed and images were recorded using a 400× inverted microscope.

### 4.10. Statistical Analyses

The mean standard deviation of three independent experiments was utilized to express all data. Tukey’s post hoc analysis was performed using SPSS 21.0 software. Statistics were deemed significant at *p* < 0.05.

## Figures and Tables

**Figure 1 molecules-28-00652-f001:**
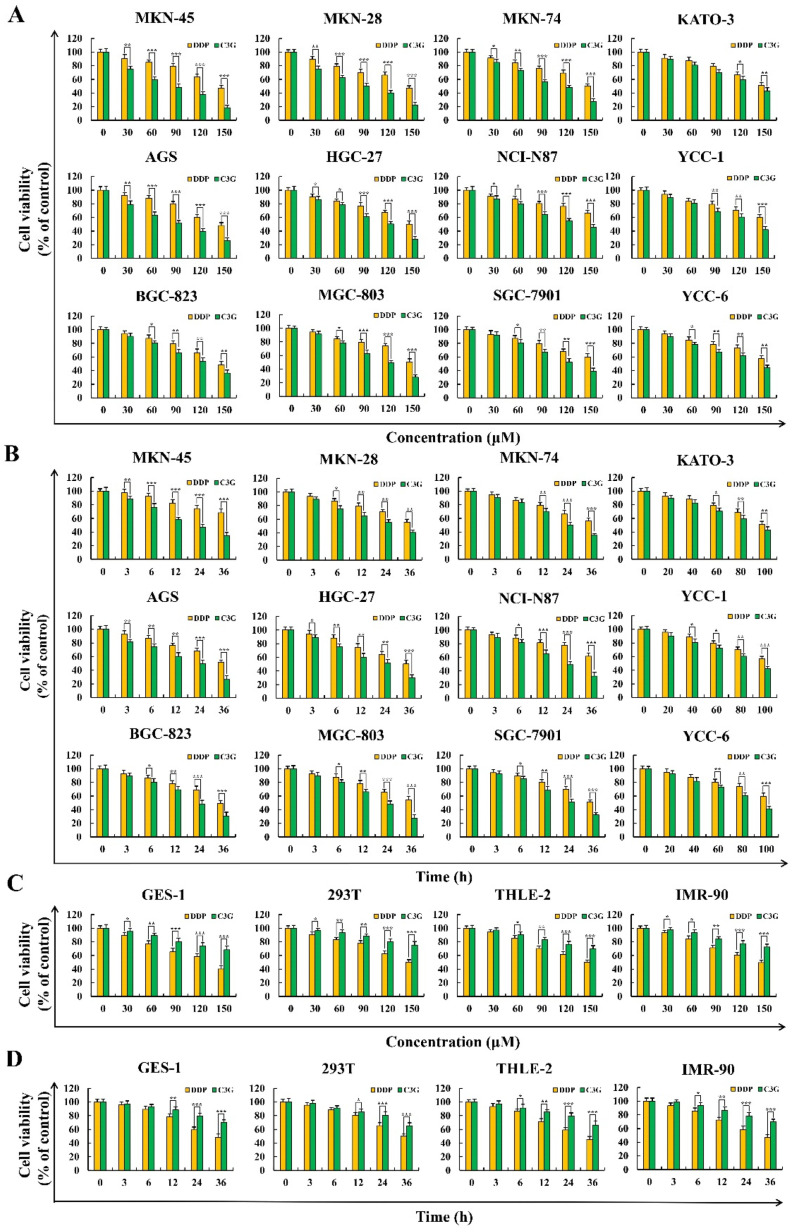
Cytotoxic effects of C3G and DDP were detected using a CCK-8 assay. (**A**) Various concentrations (30, 60, 90, 120, and 150 µM) of C3G and DDP on the toxicity of 12 human GC cell lines. (**B**) Various time points (3, 6, 12, 24, and 36 h) of C3G and DDP treatment on the toxicity of 12 human GC cell lines. (**C**) Various concentrations (30, 60, 90, 120, and 150 µM) of C3G and DDP on the toxicity of four normal cells. (**D**) Various times point (3, 6, 12, 24, and 36 h) of C3G and DDP on the toxicity of four normal cells (* *p* ≤ 0.05, ** *p* ≤ 0.01, *** *p* ≤ 0.001 vs. DDP group).

**Figure 2 molecules-28-00652-f002:**
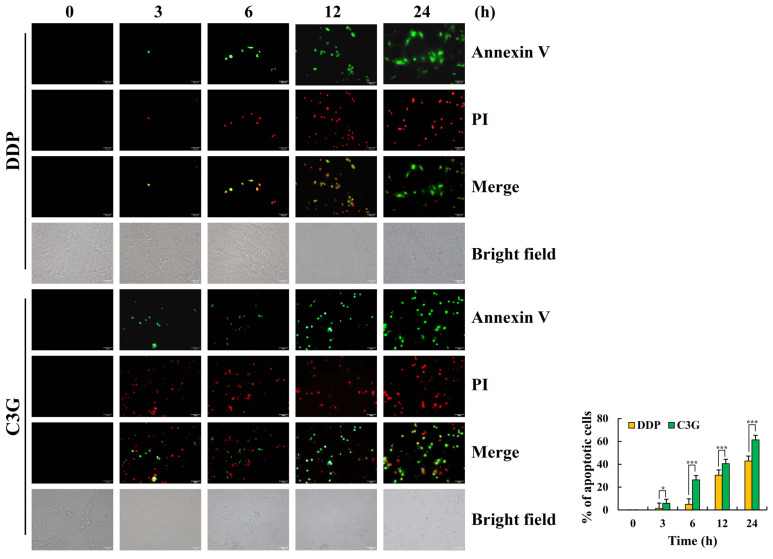
Apoptotic effects of C3G on MKN-45 cells. The fluorescence intensity and morphology of MKN-45 cells after Annexin V and PI painting were observed (original magnification 200×) (* *p* ≤ 0.05, *** *p* ≤ 0.001 vs. 0 h).

**Figure 3 molecules-28-00652-f003:**
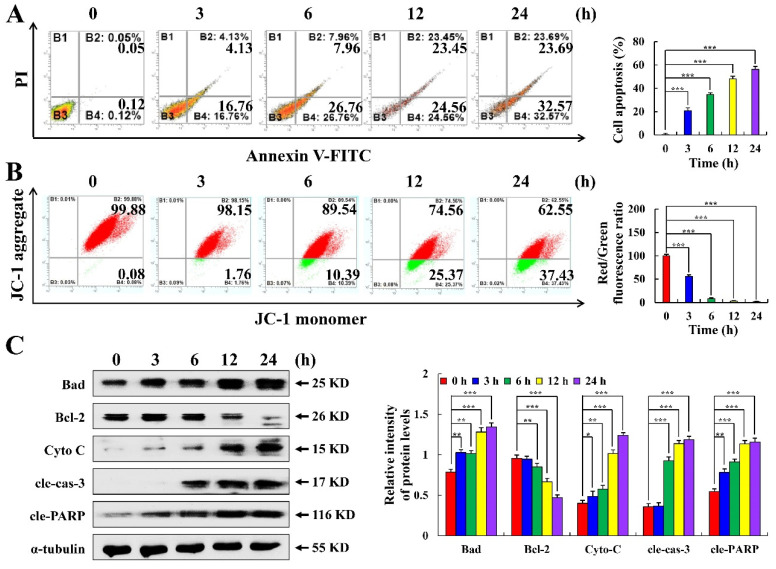
Apoptotic effects of C3G on MKN-45 cells. (**A**) The rate of apoptotic cells was determined using flow cytometry. (**B**) MMP was detected using flow cytometry. (**C**) The protein expression level of MKN-45 cells treated with C3G was detected using Western blot. α-tubulin served as an internal reference (* *p* ≤ 0.05, ** *p* ≤ 0.01, *** *p* ≤ 0.001 vs. 0 h).

**Figure 4 molecules-28-00652-f004:**
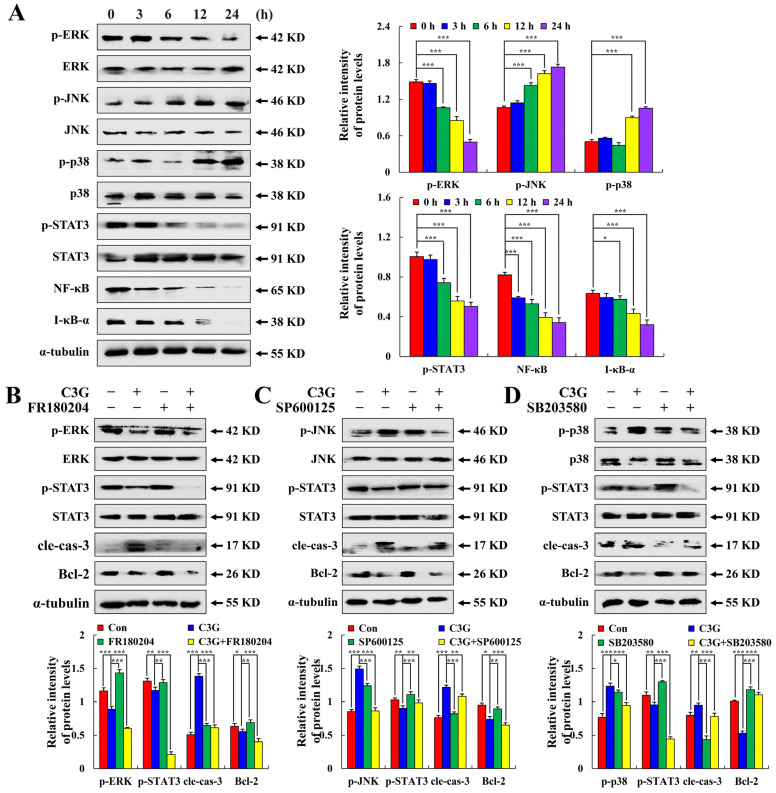
Effects of C3G on the MAPK signaling pathway in MKN-45 cells. (**A**) Western blot analysis of MAPK/STAT3/NF-κB pathway-related protein expression. (**B**) Expression levels of proteins treated with C3G and ERK inhibitors. (**C**) Expression levels of proteins treated with C3G and JNK inhibitors. (**D**) Expression levels of proteins of treated with C3G and p38 inhibitors. α-tubulin served as an internal reference (* *p* ≤ 0.05, ** *p* ≤ 0.01, *** *p* ≤ 0.001 vs. 0 h or C3G + MAPK inhibitor).

**Figure 5 molecules-28-00652-f005:**
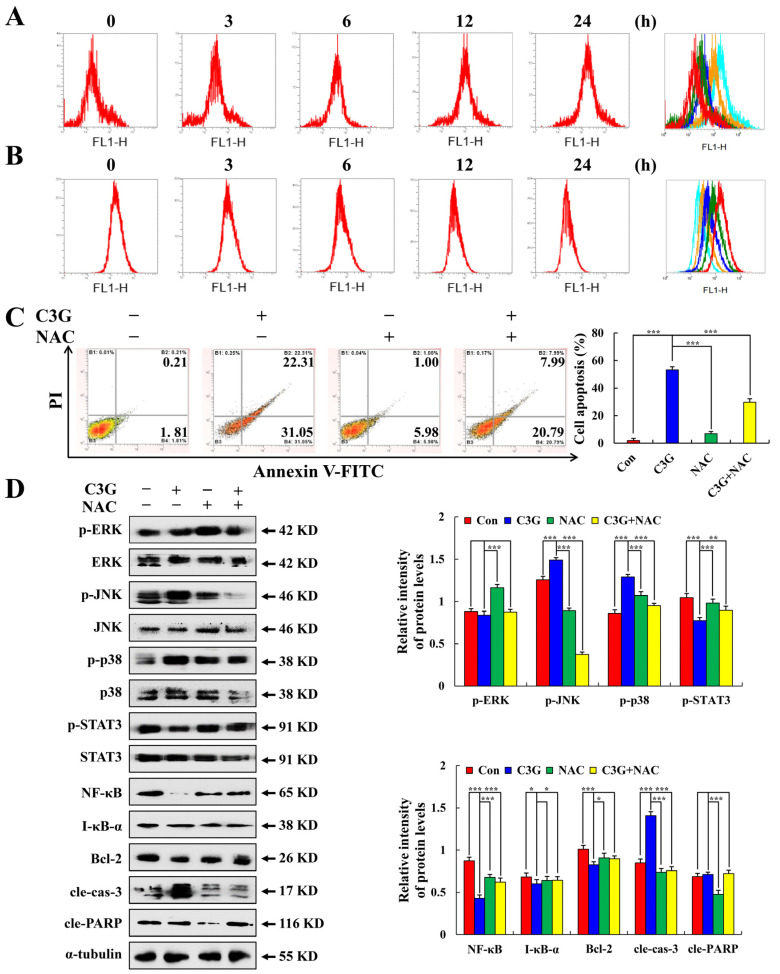
Effects of C3G on the ROS accumulation in MKN-45 cells. (**A**) ROS levels in MKN-45 cells after C3G treatment. (**B**) ROS levels in GES-1 cells after C3G treatment. (**C**) The percentage of apoptotic MKN-45 cells pretreated with NAC was detected using flow cytometry. (**D**) Western blot was used to detect the expression of apoptosis-related proteins in MKN-45 cells pretreated with NAC. α-tubulin served as an internal reference (* *p* ≤ 0.05, ** *p* ≤ 0.01, *** *p* ≤ 0.001 vs. 0 h or the NAC + C3G group).

**Figure 6 molecules-28-00652-f006:**
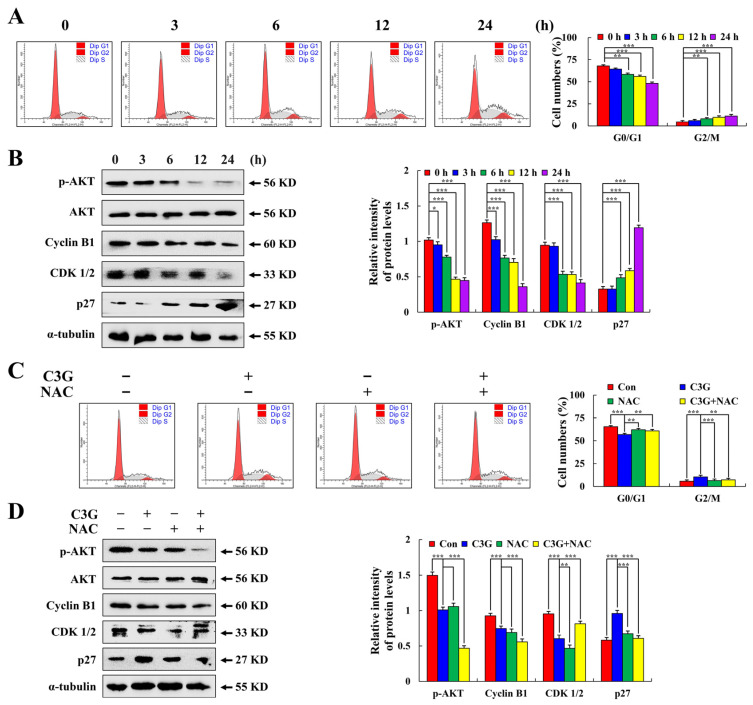
Cell cycle arrest effects of C3G on MKN-45 cells. (**A**) Percentage of cell cycle detected by flow cytometry. (**B**) G2/M cyclin-related protein expression levels after C3G treatment, cyclin AKT, cyclin B1, CDK 1/2, and p27 were detected using Western blot. (**C**) Percentage of MKN-45 cells treated with NAC and C3G was determined using flow cytometry. (**D**) MKN-45 cells were treated with C3G and NAC, and cyclin B1, CDK 1/2, and p27 were detected using Western blot. α-tubulin served as an internal reference (* *p* ≤ 0.05, ** *p* ≤ 0.01, *** *p* ≤ 0.001 vs. 0 h or the NAC + C3G group).

**Figure 7 molecules-28-00652-f007:**
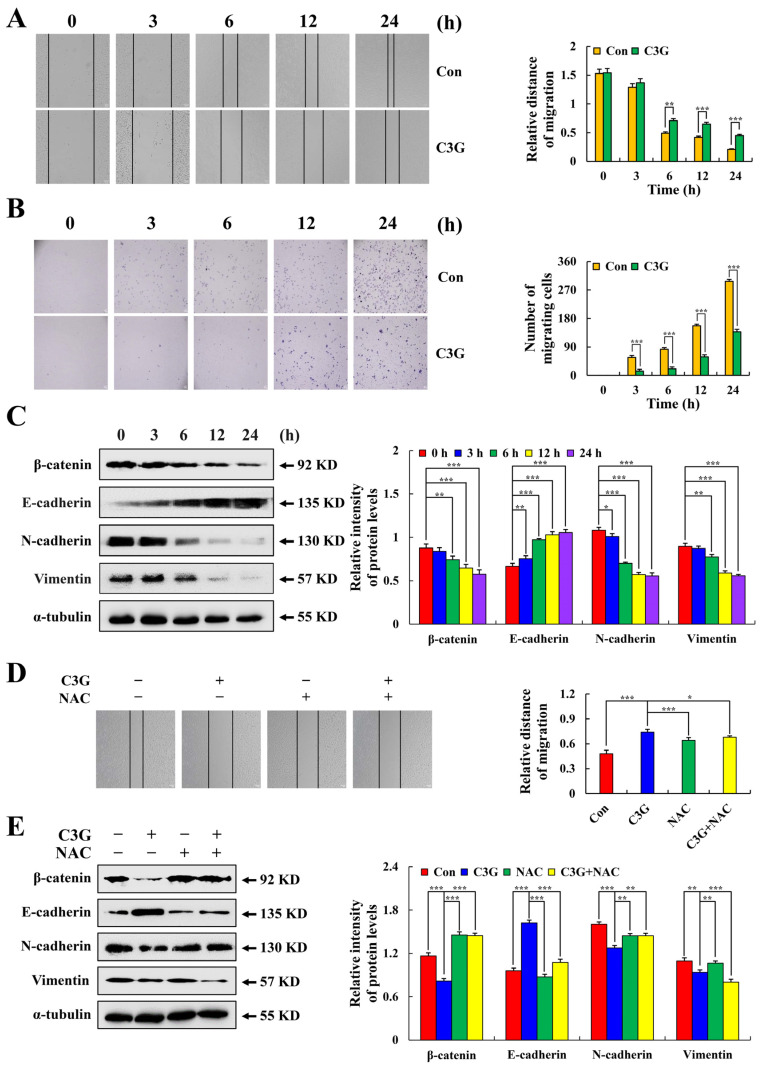
Inhibitory effect of C3G on cell migration in MKN-45. (**A**) Cell migration was observed under an inverted microscope (original magnification 100×). (**B**) Migration number of MKN-45 cells (original magnification 100×). (**C**) Western blot analysis of migration related expression levels after C3G treatment. (**D**) MKN-45 cells were treated with C3G and NAC, and cell migration was observed under an inverted microscope. (**E**) Western blot analysis of the expression of migration related expression levels after C3G and NAC treatment. α-tubulin served as an internal reference (* *p* ≤ 0.05, ** *p* ≤ 0.01, *** *p* ≤ 0.001 vs. 0 h or the NAC + C3G group).

**Figure 8 molecules-28-00652-f008:**
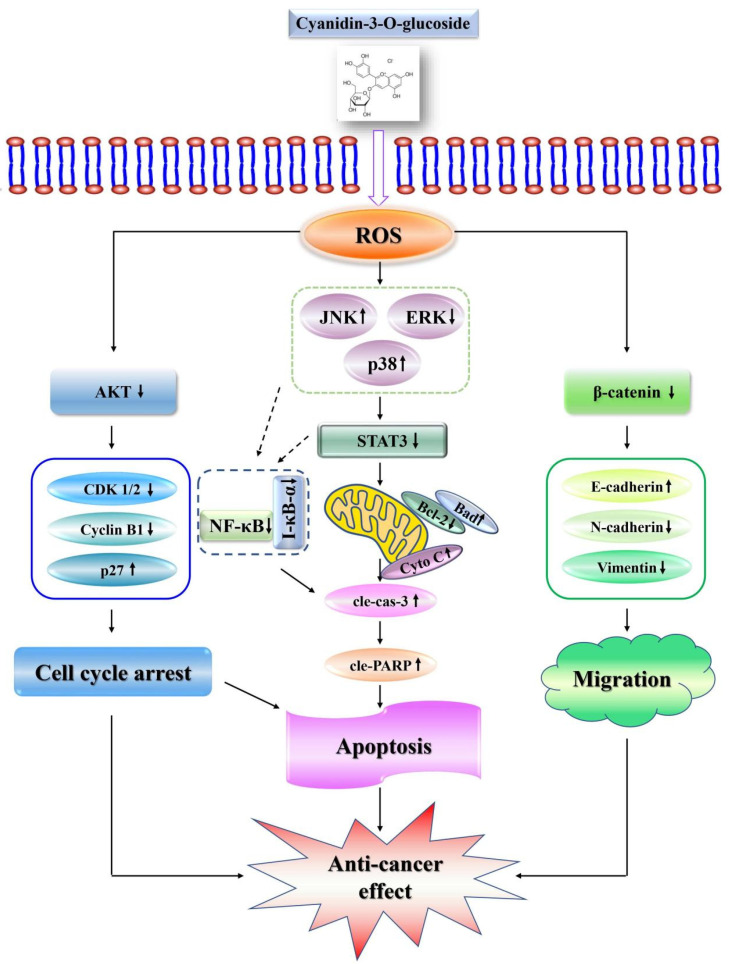
Anticancer mechanism of C3G in human GC MKN-45 cells. C3G induced apoptosis through ROS-mediated MAPK, STAT3, and NF-κB signaling pathways; arrested cells in the G2/M phase through the AKT signaling pathway; and inhibited cell migration through the β-catenin signaling pathway in human GC MKN-45 cells. Dashed arrows indicate possible regulatory relationships, and dashed boxes indicate the proteins that make up the family. The solid arrows show the relationships confirmed in this study, and the solid boxes show the proteins related to the regulation of the cell cycle and cell migration.

**Table 1 molecules-28-00652-t001:** IC_50_ values of C3G and DDP in GC cells.

Cell Line	DDP (μM)	C3G (μM)
MKN-45	157.54 ± 2.46	86.98 ± 2.46
MKN-28	150.77 ± 3.21	97.48 ± 1.84
MKN-74	153.15 ± 2.51	106.28 ± 2.17
KATO-3	152.25 ± 2.10	139.45 ± 1.38
AGS	153.02 ± 2.75	100.24 ± 2.94
HGC-27	156.78 ± 3.13	110.09 ± 3.19
NCI-N87	195.47 ± 2.41	135.69 ± 2.21
YCC-1	163.13 ± 1.56	138.45 ± 2.39
BGC-823	145.23 ± 2.18	107.80 ± 3.42
MGC-803	148.91 ± 2.48	116.08 ± 1.52
SGC-7901	160.67 ± 3.10	124.37 ± 1.34
YCC-6	157.54 ± 2.19	143.21 ± 1.84

## Data Availability

Not applicable.

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
