# Peer review of "Cyanidin-3-O-Glucoside Induces the Apoptosis of Human Gastric Cancer MKN-45 Cells through ROS-Mediated Signaling Pathways"

_molecules, 2023, doi:10.3390/molecules28020652_

Round 1
Reviewer 1 Report
In this work, the authors investigated the mechanism of tumor cell inhibition by cyanidin-3-O-glucoside, which offers important guidance for further application of the drug in vivo. However, the paper was hard to follow and it is suggested that the work be accepted after minor revision.
1. Most of words in all figures were too small to read. It was suggested that the figures were rearranged.
2. The scale bar in Figure 2 is missing and should be added.
3. What does DDP refer to? Why DDP is set as a control? These should be explained in the first paragraph of Results section.
4. What is MMP in Figure 3C legend? What is JC-1 monomer and JC-1 aggregates? What is the correlation between MMP and the JC-1 monomer/aggregates? The author should explain why the cytometry of JC-1 monomer/aggregates demonstrated the mitochondria apoptosis.
5. Explain how the ROS is measured and why the GES-1 cell is selected as the control.
6. Explain why the proteins were measured in various western blots? Why choose the proteins? The reason should be added into the manuscript.
Reviewer 2 Report
The manuscript submitted to Molecules journal investigates the anti-tumor activity of Cyanidin-3-O-glucoside (C3G) against gastric cancer cell line. The subject of the work is interesting and important due to the potential use of these preparations in gastric cancer treatment. However, this manuscript is not ready for publication yet. There are some gaps and shortcomings in the manuscript, which must be corrected before considering its publication. Detailed comments for consideration are provided below:
Introduction part:
Line31; Globally, gastric cancer (GC) is one of common cancers worldwide…..
Line 31-33; The first part is run on sentence and too complicate. Please re-phrase to improve this crucial information telling.
Line 37-38; The effects of anthocyanins should be clearly declared not only anti-cancer but also preventive ability to reduce the causes of cancer disease. More supporting evidences and reports should be summarized here.
Line 49-55; It should be the motivation or statement here about how the C3G would potentially have some effects on apoptosis, ROS accumulation, cell cycle arrest, and migration inhibition. Have this effects been investigated in other types of cancer before?
Results part:
Line 64; Since “DDP” have not been mentioned before, you should give the full name and some hypothesis or clues explaining why we have to use DDP to be compared with C3G. Could it really be considered as a “control”? and either as a positive or negative control in this particular experiment?
Figure1 and 2 are really complicated, the texts and numbers are too small to be read. The cell microscopic results should be separated as another figure. The significant modification is required to improve the quality of this cytotoxicity and apoptotic effect results.
Discussion part:
It seems like the DDP has been ignored since the introduction to the discussion part. But it just randomly pops up in the result part for no reason. You should take the DDP more serious. Introduce the general information about this compound to the readers and illustrate how you need it to be compared with C3G and how you would interpret and discuss about the results you got.
Figure 7; Please provide the definition of the dashed arrows and dashed boxes how they are different from the regular one.
